Identification and validation of a novel zinc finger protein-related gene-based prognostic model for breast cancer

Ye Min
Li Liang
Liu Donghua
Wang Qiuming
Zhang Yunuo
Zhang Jinfeng yilushunfeng80@163.com
Department of Medical Oncology 3, The Meizhou People’s Hospital , Meizhou , China
Altun Zekiye
Electronic publication date: 2021 Oct 18
Publication date: 2021
Volume: 9
Electronic Location ID: e12276
Received 2021 Jun 15; Accepted 2021 Sep 19
Copyright: © 2021 Ye et al.
Copyright year: 2021
Copyright holder: Ye et al.
License: This is an open access article distributed under the terms of the Creative Commons Attribution License, which permits unrestricted use, distribution, reproduction and adaptation in any medium and for any purpose provided that it is properly attributed. For attribution, the original author(s), title, publication source (PeerJ) and either DOI or URL of the article must be cited.
License URL: https://creativecommons.org/licenses/by/4.0/

Keywords: Bioinformatics, Breast cancer, Zinc finger proteins, Prognostic, Tumor microenvironment

Funding: The authors received no funding for this work.

==============================
Background

Breast invasive carcinoma (BRCA) is a commonly occurring malignant tumor. Zinc finger proteins (ZNFs) constitute the largest transcription factor family in the human genome and play a mechanistic role in many cancers’ development. The prognostic value of ZNFs has yet to be approached systematically for BRCA.

Methods

We analyzed the data of a training set from The Cancer Genome Atlas (TCGA) database and two validation cohort from GSE20685 and METABRIC datasets, composed of 3,231 BRCA patients. After screening the differentially expressed ZNFs, univariate Cox regression, LASSO, and multiple Cox regression analysis were performed to construct a risk-based predictive model. ESTIMATE algorithm, single-sample gene set enrichment analysis (ssGSEA), and gene set enrichment analyses (GSEA) were utilized to assess the potential relations among the tumor immune microenvironment and ZNFs in BRCA.

Results

In this study, we profiled ZNF expression in TCGA based BRCA cohort and developed a novel prognostic model based on 14 genes with ZNF relations. This model was composed of high and low-score groups for BRCA classification. Based upon Kaplan-Meier survival curves, risk-status-based prognosis illustrated significant differences. We integrated the 14 ZNF-gene signature with patient clinicopathological data for nomogram construction with accurate 1-, 3-, and 5-overall survival predictive capabilities. We then accessed the Genomics of Drug Sensitivity in Cancer database for therapeutic drug response prediction of signature-defined BRCA patient groupings for our selected TCGA population. The signature also predicts sensitivity to chemotherapeutic and molecular-targeted agents in high- and low-risk patients afflicted with BRCA. Functional analysis suggested JAK STAT, VEGF, MAPK, NOTCH TOLL-like receptor, NOD-like receptor signaling pathways, apoptosis, and cancer-based pathways could be key for ZNF-related BRCA development. Interestingly, based on the results of ESTIMATE, ssGSEA, and GSEA analysis, we elucidated that our ZNF-gene signature had pivotal regulatory effects on the tumor immune microenvironment for BRCA.

Conclusion

Our findings shed light on the potential contribution of ZNFs to the pathogenesis of BRCA and may inform clinical practice to guide individualized treatment.

Introduction

Cancer, a significant problem for global health, is the second leading cause of death in the United States (Malvezzi et al., 2019; Siegel, Miller & Jemal, 2020). The number of BRCA cases and mortality related to BRCA is globally challenging and affects several countries (Bray et al., 2018). In order to reduce the global burden that BRCA places, it is crucial to increase the effectiveness of prognostic models for overall survival (OS) so as clinical practices may be better guided (Fahad, 2019).

The Zinc-finger (ZNF) protein family, classified by small zinc finger functional domains, are able to bind: DNA, RNA, proteins, and small molecules (Li et al., 2021). Through bondage near transcription start sites and to enhancer molecules Zinc family-mediated gene expression control occurs (Hatayama & Aruga, 2018). Within the human proteome ZNF domains, found within roughly 5% of human proteins, play a role in diverse biological processes including telomere and DNA maintenance/repair, remodeling of chromatin, cell apoptosis, and autophagy (Vilas et al., 2018). Through further classification and distinction of the large ZNF protein gene class, which relate to a wide range of abnormal cell functions and instabilities within the genome such as neurodegeneration, skin disease, and diabetes (Vilas et al., 2018). ZNFs have been the subject of many studies as it has been suggested that they play a key role in tumorigenesis, cancer progression, and metastasis in a variety of cancers. Gibbs et al. (2020) reported that ZNF165, through association with SMAD 3, modulates transcription of TGFβ-dependent genes and as a result promotes triple-negative breast cancer cells. From this study it was noted that elevated ZNF165 mRNA expression negatively correlated reduced BRCA patient survival time, meaning that this specific phenotype correlated with aggressive tumorigenesis. ZNF143, found in numerous cancers including BRCA, is implied to play a mechanistic role in tumor development (Zhang et al., 2020a). Specifically, in BRCA, ZNF143 may contribute to breast cancer cell survival and recurrence-related dormancy by modulating the autophagic process (Paek et al., 2019). In previous studies, it was observed that hypermethylation of ZNF154’s promoter occurred in many tumor cell lines and gene silence was associated with a longer survival rate in resectable pancreatic cancer (Luo et al., 2018). Additionally, a recent study indicated that ZMYND8 acetylation mediates HIF-dependent breast cancer progression and metastasis (Chen et al., 2018).

Identification of biomarker signatures represents a valuable approach to mine the wealth of information contained within biological samples (Yi et al., 2020). Since the significance of ZNFs in BRCA diagnosis, treatment, and prognosis remains unclear developing a biomarker signature based on ZNF protein genes might be helpful to guide decision-making to select appropriate treatments and to predict prognosis for BRCA patients. Moreover, the prognostic performance of the signature can be enhanced by constructing nomograms that integrate, along with the gene signature and clinicopathological features (Sun et al., 2020). Recently, Zhang et al. (2021) constructed a ZNF gene-based signature based on seven genes (CLDN9, SRDA3, B4GALT1, PC, GMPPB, GPC1, B4GALT4, CHST6, and AK4) which predict OS and guide treatment in patients with bladder cancer. ZNFs role and transcription of BRCA have been studied previously however, a greater understanding of these relationships is still required.

Therefore, the goal of this study was to build a ZNF gene-based model for patient stratification, forecasting patient prognosis, and BRCA treatment guidance. Through a ZNF-gene signature and clinical variable-based nomogram, which we developed, we were able to assess the signature’s association with stromal and immune cells in the tumor immune microenvironment (TIME). Based on the expression of signature genes in low-and high-risk patients, their response to common chemotherapy agents could be predicted. Our findings shed light on the potential contribution of ZNFs to the pathogenesis of BRCA and may inform clinical practice to guide individualized treatment.

Materials & methods

Sample information and data collection

The transcriptional data and corresponding clinical information of 1066 BRCA samples and 111 normal bladder control samples were downloaded from the TCGA website (https://www.cancer.gov/tcga). Gene expression profiles were normalized by the “limma” R package. The GSE20685 dataset (Kao et al., 2011), containing mRNA expression profiles from 327 BRCA patients, was downloaded from the GEO database (https://www.ncbi.nlm.nih.gov/geo/) and used as external validation data. METABRIC dataset containing mRNA expression profiles from 1900 BRCA patients, was used as another external validation cohort. The schematic representation of the methodology of this study is shown in Fig. S1.

Construction and validation of a prognostic model

Differentially expressed genes (DEGs) between tumor and matched normal tissues were identified in the TCGA cohort via the R Package “limma” R (FDR set to <0.05). ZNF-coding genes with prognostic value were screened out by univariate Cox analysis of overall survival and p values were adjusted by Wilcoxon tests. Through Lasso’s penalized Cox regression analysis a prognostic model was created to reduce the chances of overfitting. Variable selection and shrinkage of the prognostic model were achieved by running the LASSO algorithm via R package’s “glmnet”. DEGs with prognostic value were the Models independent variables, while response variables were OS and patient status in the TCGA cohort. Result reliability and objectivity were improved via 1,000 cross-validation runs for optimal value determination of the penalty parameter (λ). Patients’ risk scores were calculated by the normalized expression level of each gene multiplied by its corresponding regression coefficient. From these values, the median value was used to stratify patients into either high- or low-risk. To evaluate the predictive power of the gene signature, a time-dependent ROC curve was built with the “survival ROC” R package. The following clinical characteristics were obtained from the TCGA database: age, stage, and tumor-node-metastasis status. These values, as well as risk scores, were pipelined into univariable and multivariate Cox regression analysis for risk-score independence determination. P values less than 0.05 were deemed statistically significant.

Using the GSE20685 and METABRIC datasets, the prognostic signature (with identical risk-score formula and threshold) was verified. Performance of the prognostic model on the validating dataset was represented via risk score-based plots depicting prognostic gene expression, risk score distribution, and survival status among patients.

Construction of a ZNF-based nomogram

Through the R package “rms”, A nomogram based on the risk score model, described prior, was constructed. Nomogram discrimination was verified via ROC analysis at 1-, 3-, and 5-year follow-up data. Predictive accuracy was assessed through a calibration plot contrasting predicted vs. actual survival.

Immunity-related modules mechanistic prediction

Stromal and immune cells play a fundamental role in shaping TIME. To further confirm the prognostic value of our signature on tumor progression, the ESTIMATE algorithm in R was used to assign stromal and immune scores to both groups defined by our constructed model (Wang et al., 2020).

Correlated pathways to our ZNF signature were explored using gene set enrichment analysis (GSEA) and a variation of it single-sample gene set enrichment analysis (ssGSEA) and gene set enrichment analysis (GSEA). Analyses of the Kyoto Encylopedia of Genes and Genomes (KEGG) were performed via the R package’s “cluster Profiler” based on the DEGs (|log2FC| ≥ 0.5, FDR < 0.05) between both groups. The Wilcoxon test was employed to adjust P-values. Within R package’s “GSVA” the ssGSEA was again used to assess tumor infiltration scores for 16 immune cell types and activation status for 13 pathways related to immunity (Hanzelmann, Castelo & Guinney, 2013).

Survival differences among HNSCC subtypes were assessed using the Java program GSEA, using Hallmark gene set “h.all.v7.0.symbols.gmt” from the MSigDB, with a 1,000 permutation random sampling. A two subtypes enrichment pathway was determined via a false discovery rate (FDR) of <0.05 and NES.

Prediction of chemotherapeutic and molecular-targeted therapy response

The chemotherapeutic response of each BRCA patient in the TCGA cohort was predicted according to the pharmacogenomic database Genomics of Drug Sensitivity in Cancer (GDSC), publicly available at: https://www.cancerrxgene.org/. The GDSC database contains data from a large collection of human cancer cell lines, anticancer compounds, and experimental data on drug sensitivity. The prediction of drug sensitivity (IC50) values was conducted using the R package “pRRophetic”, which uses a ridge regression model based on cancer cell lines’ expression profiles in the GDSC (Yang et al., 2012).

Results

Identification of differentially expressed ZNFs-related genes in BRCA

To determine the expression patterns of ZNFs-related genes in BRCA, expression levels of 1,818 human ZNFs protein-coding genes retrieved from the UniProt database (Table S1) were evaluated in the transcriptional profiles of 1,066 BRCA samples and 111 normal breast samples, available in the TCGA. A total of 171 upregulated and 149 downregulated ZNF-coding, differentially expressed genes (DEGs) were thus identified (Figs. 1A, 1B and Table S2).

Figure 1 Identification of DEGs of ZNFs in the TCGA-BRCA dataset.

(A) Heatmap depicting the expression levels of ZNFs-related genes in BRCA (T) and normal (N) samples. (B) Volcano plot representation of DEGs of ZNFs.

Construction of a ZNFs-related gene-based risk signature for BRCA

A total of 998 patients, with a >30-day follow-up time, were selected from the TCGA-BRCA dataset for signature construction. Through Univariate Cox regression, 28 prognostic-associated candidate ZNFs-related genes were obtained, from 320 potential candidates (Fig. 2A, all P < 0.05; Table S3). A total of 14 significant DEGs (ZMYND10, SIAH2, ZNF239, ZNF219, TANK, WT1, FGD5, PARP12, SHARPIN, ZBED2, ZBED3, MECOM, OVOL1, and IKZF3), filtered via LASSO and multivariate Cox regression, were significantly independently correlated with OS and prognostic value (adjusted P < 0.05). From these 14 ZNF genes, a signature was built allowing for BRCA survival risk evaluation (Fig. 2B and Fig. S2).

Figure 2 Univariate and multivariate Cox regression analysis of the TCGA-BRCA dataset illustrated as a forest plot.

(A) Univariate Cox results of differential ZNFs illustrated by forest plot. (B) Forest plot showing prognostic ZNFs-related genes in BRCA based on multivariate Cox results. * P < 0.05, ** P < 0.01, *** P < 0.001.

Patients’ risk scores were calculated in the training set based on the 14-gene signature. The median cut-off value from these risk-scores in the TCGA-BRCA dataset was used to stratify patients into groups, high-risk and low-risk (both n = 499) (Fig. 3A). As shown in Fig. 3B, high-risk patients were more likely to have an early death than low-risk patients. Consistently, the heatmap of expression profiles in the TCGA-BRCA dataset showed distinct differences between groups (Fig. 3C and Fig. S3). As shown in Fig. S3, among the 14 ZNF-related prognostic genes, the expression levels of MECOM, OVOL1, WT1, ZBED2, ZBED3 and ZNF239 were significantly higher in the high-risk group, whereas the expression levels of IKZF3, PARP12, SIAH2, TANK, ZMYND10, and ZNF219 were significantly higher in the low-risk group. Through survival analysis, it was observed that high-risk patients had a significantly poorer OS than low-risk patients (Fig. 3D, P < 0.001). Further, we performed prognostic analysis of breast cancer subtypes, and the results indicated that the 14 ZNFs gene-based signature shows good prognostic value in triple negative breast cancer (TNBC) and non-TNBC (Fig. S4). AUC values for our observed overall survival groups (1-, 3- and 5-years) and the 14 ZNFs gene-based prognostic signature were 0.811, 0.638, and 0.639, respectively, illustrating adequate performance (Fig. 3E).

Figure 3 Development of a prognostic signature for BRCA based on 14 ZNFs-related genes.

(A) Risk score distribution and median value in the TCGA-BRCA cohort. (B) Patient survival status in differing risk groups. (C) Expression profile heatmap of the 14 ZNFs-related gene prognostic signature. (D) Signature-defined risk group survival analysis. (E) Time-dependent ROC curve for prognostic 14-gene signature.

External prognostic signature validation

Our 14 ZNFs gene-based signature prognostic value was verified using patient data from the GSE20685 dataset. This served as external testing after high- and low-risk group categorization via calculated median and cutoff value for the model (Fig. 4A). The expression profiles corresponding to the signature genes are shown in Fig. 4C. Similar to that in the previous analysis, the high-risk patient group was associated with earlier death, (Fig. 4B) and reduced overall survival time in comparison to the low-risk patients (Fig. 4D; P < 0.001). Similar results were observed in METABRIC dataset (Fig. S5, P = 0.023). Within the GSE20685 population the 14 ZNF gene-based signature prognostic capabilities showed acceptable discrimination, which is illustrated by AUCs values of 0.732, 0.768, and 0.737, for 1-, 3-, and 5-year OS, respectively (Fig. 4E).

Figure 4 Validation of the 14-ZNF-gene prognostic signature in GSE20685 dataset.

(A) Population risk score distribution and the median value for GSE20685. (B) Survival status of low-risk and high-risk patients. (C) The 14-ZNF-gene signature expression profile heatmap. (D) Signature-defined risk group survival analysis. (E) Prognostic signature’s time-dependent ROC curve.

Establishment of a ZNF gene-based nomogram

Univariate analyses were performed to examine the prognostic values of several clinicopathological features (age, tumor stage, T stage, N stage, M stage, and risk score). Consequently, the 14-ZNFs risk signature was associated with overall survival (hazard ratio [HR] = 1.154; 95% confidence interval [CI], [1.120−1.189], P < 0.001) (Fig. 5A). BRCA risk factors also include being older than 56 years, T stage, M stage, and N stage (Fig. 5A). Through multivariate analyses, it was revealed that risk score (HR = 1.125; 95% CI, [1.091−1.160], P < 0.001) and age >56 (HR = 1.551; 95% CI, [1.073−2.241], P = 0.020) remained independent prognostic factors (Fig. 5B). The 14-ZNFs risk signature had greater sensitivity and specificity (AUC = 0.747) than other clinicopathological features as illustrated by ROC curve analysis (Fig. 5C). This information was pipelined along with patient clinicopathological data for nomogram construction 1-, 3-, and 5-year OS forecasting (Fig. 5D). The C-index (Fig. 5E) and calibration plot (Fig. S6) of the nomogram indicated optimal predictive accuracy, with a close overlap between predicted and actual survival rates.

Figure 5 14-ZNF-gene signature in TCGA-BRCA dataset based nomogram.

(A) Univariable and (B) multivariable analyses adjusting for risk score, age, tumor stage, and TNM stage. (C) ROC curve respecting clinical features and risk model. (D) Predictive nomogram for OS in BRCA patients. (E) C-indes of the nomogram.

The 14 ZNF-gene signature predicts differences in TIME of BRCA

To assess whether the 14 ZNF-gene signature can help distinguish differences in the tumor microenvironment of BRCA, we employed the ESTIMATE algorithm to compare gene expression signatures of stromal and immune cells among risk groups. The stromal score ranged from −2,065.58 to 2,109.48 (Fig. 6A, P < 0.001), the immune score ranged from −1,182.02 to 3,672.57 (Fig. 6B), and the ESTIMATE score ranged from −2,916.86 to 5,355.63 (Fig. 6C). Compared with the low-risk group, the high-risk group had a significantly lower Immune Score (Fig. 6B, P < 0.001) and ESTIMATE Score (Fig. 6C, P < 0.001). From this, we can determine that high-risk patients had lower immune cell infiltration. It can also be stated that our ZNF-gene signature may be a procrastinator for TIME status.

Figure 6 Tumor microenvironment composition group comparison among the TCGA-BRCA dataset.

(A) Comparison of stromal scores between risk groups. (B) Comparison of immune scores between risk groups. (C) Comparison of ESTIMATE scores between risk groups. *** P < 0.001.

Correlation between immune infiltrations and functions and the ZNF-gene signature

To further investigate the 14 ZNF-gene signature’s risk score relations the following were quantified via ssGSEA within the TCGA-BRCA database: tumor immune status, diverse immune cell subpopulations’ enrichment scores, and their related functions or pathways. The risk score and infiltration level relationships for 16 immune cell types (dendritic cells (DCs), activated DCs (aDCs), plasmacytoid DCs (pDCs), iDCs, B cells, CD8 + T cells, T helper cells, T follicular helper cells (Tfhs), Th1 cells, Th2 cells, tumor-infiltrating lymphocytes (TILs), regulatory T cells (Tregs), Neutrophils, NK cells Macrophages and Mast cells were analyzed in order to estimate the 14 ZNF-gene signatures effect on BRCA TIME. Risk-score and infiltration levels of the 14 observed immune cell types were negatively correlated significantly (Fig. 7A, P < 0.05 for all). On immune function analysis, the low-risk group showed activity enrichment in cytolytic activity, inflammation-promoting, parainflammation, and higher scores for checkpoint molecules, HLA, MHC class I, type I, and type II IFN responses (Fig. 7B, P < 0.01 for all). This result further indicated that our 14 ZNF-gene risk signature was implicated in the TIME of BRCA.

Figure 7 Risk group ssGSEA score comparison in the TCGA-BRCA dataset.

(A) A total of 16 immune cell types scores. (B) Functions enriched in the 14-ZNF-gene signature. ** P < 0.01, *** P < 0.001.

The 14 ZNF-gene signature predicts chemotherapy and targeted therapy response in BRCA

Considering that chemotherapy is still the most effective adjuvant measure to treat BRCA, we accessed the Genomics of Drug Sensitivity in Cancer (GDSC) database to estimate the response of low-risk and high-risk BRCA patients to commonly used drugs. The correlation between risk groups and IC50 values for 138 chemotherapeutic agents was visualized using scatterplots. We found significant discrimination between groups in the estimated IC50 values of six common chemotherapy drugs (gemcitabine, vinblastine, vinorelbine, cisplatin, docetaxel, and doxorubicin; Fig. 8A, P < 0.05 for all) and three molecular-targeted drugs (erlotinib, gefitinib, and sunitinib; Fig. 8B, P < 0.05 for all) used in BRCA treatment. The estimated IC50 values of these drugs except docetaxel were significantly elevated in high-risk samples of the TCGA-BRCA dataset (Fig. 8).

Figure 8 Chemotherapy therapy and molecular-targeted therapy response predictions for TCGA-BRCA risk groups.

(A) Boxplots exhibiting the estimated IC50 values of six common chemotherapy drugs for tumor cells from the two risk groups. (B) Boxplots exhibiting the estimated IC50 values of three molecular-targeted drugs for tumor cells from the two risk groups. *** P < 0.001.

Signaling pathways analysis of the ZNF-related signature in BRCA

To gain insight into the functions of the 14 ZNF protein-coding genes included in our signature, we performed KEGG enrichment analysis based on GSEA enrichment scores. The results indicated that the expression patterns that conformed to the low-risk group were enriched in KEGG terms related to tumor progression, such as JAK STAT, VEGF, MAPK, NOTCH TOLL-like receptor, NOD-like receptor signaling pathways, apoptosis, and pathways in cancer (Fig. 9A and Table S4, FDR q-val < 0.05 for all). Notably, our ZNF-gene signature was closely correlated with were mainly enriched in immune-related signaling pathways such as cytokine-cytokine receptor interaction, natural killer cell-mediated cytotoxicity, antigen processing and presentation, chemokine signaling pathway, T cell and B cell receptor signaling pathways (Fig. 9B and Table S4, FDR q-val < 0.01 for all). These results suggest that the ZNF protein genes comprising our BRCA signature may also drive the onset or progression of cancers by modulating the tumor immune microenvironment.

Figure 9 GSEA of the ZNF-related signature in BRCA.

(A) Pathways related to tumor progression. (B) Pathways related to TIME.

Discussion

BRCA, has the number one cancer-related mortality among women worldwide (DeSantis et al., 2019), due to its phenotypic and molecular diversity is difficult to accurately predict disease prognosis. Prognostic model application is essential for clinical decision guiding and precision medicine. Current prognostic models have flaws due to subtype misidentification, inadequate risk stratification, and lack of underlying mechanism characterization. Fixing these issues will allow for precise and personalized therapies and prolongation of survival time (Zhang et al., 2021).

Here we illustrated expression patterns, prognosticator capabilities, and TIME effects of ZNF related genes in BRCA. We performed Univariate and multivariate Cox regression analysis, as well as LASSO regression analyses to identify 14 ZNFs-genes with prognostic capabilities for BRCA patients from the TCGA dataset. From these 14 genes, an effective model was developed and tested for clinical outcome prediction in BRCA patients. Based upon group survival analysis, there were distinct prognoses differences among BRCA patients. As the 14 ZNF-genes model, showed capable predictive abilities within the GSE20685 and METABRIC datasets (used for verification), it could be useful in the clinical field as a predictive model. Through multivariate Cox analyses, the 14 ZNF-genes model could act as an independent prognosticator for BRCA patients. Comparatively, our current model outperformed traditional clinical factors including age and TNM stage. Based on the complementary values of clinical characteristics and our 14 ZNF-genes prediction model a novel nomogram was generated that provides superior OS estimation for BRCA patients. Our nomogram performed well when analyzed with ROC curve, C-index analysis and calibration plot for 3- and 5-year subgroupings. The results indicated that our 14 ZNF-genes model is competitive to traditional prognostic survival prediction models for BRCA.

The 14 genes of our ZNFs-related signature identified in this study included: ZMYND10, SIAH2, ZNF239, ZNF219, TANK, WT1, FGD5, PARP12, SHARPIN, ZBED2, ZBED3, MECOM, OVOL1, and IKZF3. Among these, six genes (ZMYND10, SIAH2, SHARPIN, FGD5, WT1, and OVOL1) have been implicated, as discussed below, in the tumorigenicity and progression of BRCA. For example, ZMYND10 (zinc finger, MYND type containing 10) has been labeled as a gene involved possibly in tumor suppression, making it capable of cell cycle arrest, proliferation and angiogenesis inhibition, as well as apoptosis induction in multiple tumor types (Wang et al., 2019; Cheng et al., 2015). ZMYND10 has been observed, by recent studies, to sensitize anticancer activities of chemotherapeutic agents (gemcitabine (Yoo et al., 2013) and paclitaxel (Park et al., 2013) to name a few). Wang et al. (2019) revealed that ZMYND10 plays an inhibitory role within the miR145-5p/NEDD9 signaling pathway, this ultimately leads to suppression of BRCA tumorigenesis. SIAH2, an E3 ubiquitin ligase, has key functions in multiple fundamental cellular processes: hypoxia, intracellular signaling, and the unfolded protein response to name a few (Scortegagna et al., 2020; Ma et al., 2019). SIAH2 protein expression was shown to be able to predict ER status and tamoxifen therapy outcome due to its inverse relationship within a metastatic BRCA cohort (van der Willik et al., 2016). SIAH2-NRF1 axis remodels tumor microenvironment by modulating tumor mitochondrial function, tumor-associated macrophage polarization, and cell death for tumorigenesis and progression in BRCA (Ma et al., 2019). SHanK-associated rH domain-interacting protein (SHARPIN), an atypical ubiquitin-binding protein, was observed to promote BRCA progression when highly expressed and regulates cancer development for a series of other tumors (Tian et al., 2019). SHARPIN does have key roles in carcinogenic pathway control in BRCA, such as ERα and P53 where it inhibits protein stability via MD32. Additionally, SHARPIN was identified in vivo as a BRCA metastasis gene and predicts metastasis-free survival after adjuvant therapy (Bii, Rae & Trobridge, 2015). FGD5 (Faciogenital Dysplasia 5) amplification in BRCA is associated with higher tumor proliferation and poorer OS (Valla et al., 2017). Wilms’ tumor gene 1 (WT1) is expressed highly in BRCA; as a result, vaccines limiting WT1 expression are in phase 1 and phase 2 of clinical trials, however, side effects and efficacy are still under question (Zhang et al., 2020b). OVO-like proteins 1 (OVOL1) can induce metastasis in prostate and BRCA cells characterized by regulating mesenchymal to epithelial transition (Roca et al., 2013; Saxena et al., 2020b). In contrast, little research has been done on the roles of ZBED2, ZBED3, ZNF219, TANK, IKZF3, PARP12, ZNF239, and MECOM in BRCA onset and development, meaning more research on their biological functions in regards to BRCA is warranted.

Interestingly, we found that high expression of MECOM was significantly associated with advanced TNM stage (Fig. S7B, P < 0.001), T stage (Fig. S7C, P = 0.044), and N stage (Fig. S7D, P < 0.001) in BRCA patients. Therefore, we identified MECOM as target for clinical and experimental analyses in the future. As shown in Fig. S8, among the 14 ZNF-related prognostic genes, IKZF3, PARP12, TANK, WT1, and ZBED2 were upregulated, while FGD5, MECOM, SIAH2, and ZMYND10 were down-regulated in TNBC compared with non-TNBC based on TCGA and METABRIC datasets, suggesting these genes are associated with TNBC. In addition, the correlations between these ZNF-related prognostic genes are presented in Fig. S9. The interactions among the 14 genes, as well as the specific biological functions of these genes in BRCA need to be explored experimentally.

TIME played a critical role in the initiation and progression of tumorigenesis. Based on the results of ESTIMATE, ssGSEA, and GSEA analysis, we elucidated that our ZNF-gene signature had pivotal regulatory effects on the TIME in BRCA. Through ESTIMATE algorithm calculation, we illustrated low-risk patients had significantly higher immune scores in comparison to high-risk patients. Furthermore, we evaluated the correlation between the patterns and tumor immune microenvironment features using ssGSEA. Our data showed vast differences in immune cell infiltration and function between the risk groups, especially for T cells, B cells, dendritic cells (DCs), and NK cells. It has been previously established that greater immune cell infiltration levels were positively correlated with immunotherapy responsiveness in multiple tumor types (Karn et al., 2017; Kümpers et al., 2019). Some previous studies have separated BRCA into two different subgroups based on immune cell infiltration: “hot” (enriched in immune cells infiltration) and “cold” (lack of immune cells infiltration) which served as a strong indicator for immunotherapy response (Sharma & Allison, 2015; Cheng et al., 2018). In a similar manner, our ZNF-gene signature also has potential clinical value by utilizing immune cell infiltration and immune function in predicting immunotherapy responses. GSEA was employed for the investigation of mechanisms underlying the 14 ZNF protein-coding genes in BRCA. The ZNF-gene signature was closely correlated with mechanisms enriched in: cytokine-cytokine receptor interaction, natural killer cell-mediated cytotoxicity, antigen processing/presentation, chemokine, T cell receptor, and B cell receptor signaling pathways. In addition, the low-risk subgroup was enriched with JAK STAT, VEGF, MAPK, NOTCH, NOD-like and TOLL-like receptor signaling pathways, apoptosis, and pathways in cancer, which were also important oncogene targets in cancer development. Prior to this analysis, these hallmarks were already recognized to be related to immune reaction, cancer progression, and cancer immunotherapy.

Chemotherapy and targeted therapy are still important ways of cancer treatment. By analyzing the GDSC database, we found that the high-risk group and the low-risk group had different sensitivities to six common chemotherapy drugs, of which gemcitabine, vinblastine, vinorelbine, cisplatin, and doxorubicin were relatively sensitive within low-risk patients, while docetaxel seemed more resistant. Interestingly, our risk model had an effect on cell sensitivity to three molecular-targeted drugs, such as erlotinib, gefitinib, and sunitinib. This evidence suggests that our risk model may provide a reference for the treatment choice in BRCA patients.

As far as we are aware, this study is the first to comprehensively analyze and identify ZNFs as prognosticators for survival in BRCA patients via data evaluation from the TCGA and GEO databases. From this analysis, a 14 ZNF-related gene-based risk signature was constructed and tested. This model is capable of stratifying BRCA patients for increased effectiveness of immune and chemotherapy. A ZNF-related nomogram integrating our model and incorporating clinical factors and molecular markers capable of OS prediction for BRCA patients was established. Through this study, we were able to shine a light on the genetics behind BRCA as well as significantly guide future research.

Our study was limited by the following: (1) As a retrospective study and focused solely on BRCA. (2) Large cohorts are required for valid predictive performance while using this model, (3) Its clinical applicability still needs to be validated for better management of BRCA, (4) Basic experiments must be completed in the future to verify findings and shine more light on ZNFs mechanistic role within tumorigenesis and development of BRCA, (5) Our current gene-signature model may not be as effective on patients who are not affected by distal metastasis and needs further prognostic testing.

Conclusions

In summary, we established a novel 14 ZNFs gene-based signature prognosticator that divides BRCA patients into high- and low-risk subgroups which are characterized by differing survival outcomes, and constructed a nomogram to help clinical decision-makers provide optimal treatment. The prognostic signature is associated with immune cell components and functionality differences within tumor microenvironments. The signature also predicts sensitivity to chemotherapeutic and molecular-targeted agents in high- and low-risk patients afflicted with BRCA. Our study may stimulate further research on the role of ZNFs on BRCA and help guide stratified therapy to provide individualized treatment. For a better understanding of these results, and ZNFs mechanistic relation to BRCA more research must be completed.

Supplemental Information

Supplemental Information 1 1818 ZNFs-related genes.

Click here for additional data file.

Supplemental Information 2 The 320 DEGs of ZNFs in BRCA.

Click here for additional data file.

Supplemental Information 3 Unicox results of differential ZNFs-related genes.

Click here for additional data file.

Supplemental Information 4 GSEA of the ZNF-related signature in BRCA.

Click here for additional data file.

Supplemental Information 5 Study design and workflow of this study.

Click here for additional data file.

Supplemental Information 6 The LASSO Cox regression model was used to identify the most robust markers.

(A) Ten-fold cross-validation for the coefficients of 320 candidate ZNFs-related genes in the LASSO model. (B) X-tile analysis of the 22 selected ZNFs-related genes.

Click here for additional data file.

Supplemental Information 7 Differential expression of the 14 ZNF-related prognostic genes between high-and low-risk group in TCGA-BRCA dataset.

Click here for additional data file.

Supplemental Information 8 OS of the 14 ZNFs gene-based signature in TNBC and non-TNBC based on TCGA-BRCA dataset.

Click here for additional data file.

Supplemental Information 9 OS of the 14 ZNFs gene-based signature in breast cancer based on METABRIC dataset.

Click here for additional data file.

Supplemental Information 10 The 1-, 3-and 5-year nomogram calibration curves.

Click here for additional data file.

Supplemental Information 11 Association between expression levels of the 14 ZNF-related prognostic genes and clinical features in TCGA-BRCA dataset.

Click here for additional data file.

Supplemental Information 12 Differential expression of the 14 ZNF-related prognostic genes between TNBC and non-TNBC.

Click here for additional data file.

Supplemental Information 13 The interaction between 14 ZNF-related prognostic genes in TCGA-BRCA dataset.

Click here for additional data file.

Supplemental Information 14 R scripts.

Click here for additional data file.

Additional Information and Declarations

Competing Interests

Author Contributions

Data Availability

The authors declare that they have no competing interests.

Min Ye conceived and designed the experiments, performed the experiments, analyzed the data, prepared figures and/or tables, authored or reviewed drafts of the paper, and approved the final draft.

Liang Li conceived and designed the experiments, performed the experiments, analyzed the data, authored or reviewed drafts of the paper, and approved the final draft.

Donghua Liu analyzed the data, authored or reviewed drafts of the paper, and approved the final draft.

Qiuming Wang conceived and designed the experiments, prepared figures and/or tables, and approved the final draft.

Yunuo Zhang analyzed the data, prepared figures and/or tables, and approved the final draft.

Jinfeng Zhang conceived and designed the experiments, authored or reviewed drafts of the paper, and approved the final draft.

The following information was supplied regarding data availability:

The raw measurements are available in the Supplementary Files.

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
