# Peer review of "Identification and validation of a novel zinc finger protein-related gene-based prognostic model for breast cancer"

_PeerJ, doi:10.7717/peerj.12276_

## Round 0.1 · original submission · Major Revisions

The manuscript is related to zinc finger proteins in breast cancer. The authors have focused on 14-genes as a signature for breast cancer. The authors must consider the reviewer's suggestions.

Reviewer 1 ·

Basic reporting

Clearly written

Experimental design

Reasonable

Validity of the findings

Mostly fine

Additional comments

This study explores the clinical significance of the expression profiles of Zinc finger proteins in breast cancer. The authors identified 14 Zinc finger proteins that are differentially expressed in breast tumors as compared to normal cells and associated with patient survival. The authors focused on these 14 genes and generated a gene signature. This newly built signature is associated with patient prognosis, immune score, and drug sensitivity. Overall, the manuscript is carefully written, and the detailed discussion of the identified 14 genes is very valuable for future studies. Particularly, the identification of 8 zinc finger proteins (ZBED2, ZBED3, ZNF219, TANK, IKZF3, PARP12, ZNF239, and MECOM) that are not appreciated before is an important finding in this paper.

However, the authors heavily focused on the 14-gene signature and most of the findings are descriptive. Breast cancer subtypes are well established, and it is not clear how the 14-gene signature is unique and what aspects are better. Although the authors confirmed the prognostic value with another data cohort, it would be important to confirm such a prognostic value with other large data cohort such as METABRIC.

The expression profile of the 14 ZNFs-related gene prognostic signature (Figure 3C) is not clearly showing the differential expression profiles among these 14 zinc finger proteins. This also suggests that 14 genes are not necessary to build such a gene signature. Since these 14 genes are independently associated with prognosis, it would be more interesting to see common and distinct clinical features among these 14 genes based on the expression or mutations. For instance, how frequently are they co-expressed? What kind of breast tumor subtypes are associated with these genes?

It is also unclear why one of the well-known zinc finger proteins in breast cancer, GATA3, is excluded from the analysis. GATA3 expression is known to be correlated with patient survival.

Minor comments

Line 168 “Have an early than low-risk patients”
Please clarify this statement.

Figure 1 B: Since many plots are hidden, it would look better if the scale of X-axis is adjusted.

Reviewer 2 ·

Basic reporting

The article is well written and the methodology is well presented. I would suggest a schematic representation of the methodology since the mere description is difficult to grasp. I got a single sentence that is not clear: line 168, apparently, a word is missing.

The literature references are comprehensive and adequate.

The article structure, figures, and tables are clear and relevant. The article used publicly available raw data. But the generated data are well presented and completed. I would suggest the authors to share the R scripts used in the article.

Experimental design

The article is within the journal's scope.

The question is well defined, is relevant, and meaningful. The question is interesting and has clinical applications and also is relevant from a basic research perspective.

The investigation used a well-supported methodology and follow ethical standards.

The methods are well described with reasonable details. I am suggesting the inclusion of the R scripts to guarantee replication.

Validity of the findings

The article proposes a score based on zinc fingers proteins as a predictive tool to use on clinical set to detect patient OS and drug response. The justification of the proposed score is not justified, but it seems to work quite well on this problem. The methodology is adequate and the discussion is well presented.

The basic data is public and the results are reasonably robust, although some AUC values are below 0.7.

The conclusions are well supported by the results and answer the stated problem.

Additional comments

In my opinion, the article presents interesting results that could be used in other clinical contexts. If the authors could furnish the scripts this could evolve to a powerful workflow to be applied to other cancer types.

---

## Round 0.2 · accepted · Accept

The authors have been done all required changes in the manuscript and validations.

Reviewer 1 ·

Basic reporting

Clearly written

Experimental design

Well-considered

Validity of the findings

Sufficient validation was performed

Additional comments

The authors addressed all concerns.